# 3D nitrogen-doped graphene foam with encapsulated germanium/nitrogen-doped graphene yolk-shell nanoarchitecture for high-performance flexible Li-ion battery

Runwei Mo[1,2], David Rooney[3], Kening Sun[2] & Hui Ying Yang[1]

Flexible electrochemical energy storage devices have attracted extensive attention as promising power sources for the ever-growing field of flexible and wearable electronic products. However, the rational design of a novel electrode structure with a good flexibility, high capacity, fast charge–discharge rate and long cycling lifetimes remains a long-standing challenge for developing next-generation flexible energy-storage materials. Herein, we develop a facile and general approach to three-dimensional (3D) interconnected porous nitrogen-doped graphene foam with encapsulated Ge quantum dot/nitrogen-doped graphene yolk-shell nano architecture for high specific reversible capacity ($1,220\,\mathrm{mAh\,g^{-1}}$), long cycling capability (over 96% reversible capacity retention from the second to 1,000 cycles) and ultra-high rate performance (over $800\,\mathrm{mAh\,g^{-1}}$ at 40 C). This work paves a way to develop the 3D interconnected graphene-based high-capacity electrode material systems, particularly those that suffer from huge volume expansion, for the future development of high-performance flexible energy storage systems.

[1] Pillar of Engineering Product Development, Singapore University of Technology and Design, 8 Somapah Road, Singapore 487372, Singapore. [2] Academy of Fundamental and Interdisciplinary Sciences, Harbin Institute of Technology, Harbin 150001, China. [3] School of Chemistry and Chemical Engineering, Queen's University Belfast, Belfast BT9 5AG, Northern Ireland. Correspondence and requests for materials should be addressed to K.S. (email: keningsunhit@126.com) or to H.Y.Y. (email: yanghuiying@sutd.edu.sg).

The ever-growing demands in future flexible electronics, such as wearable devices, portable electronic devices and implantable biomedical products, have aroused worldwide research interests in the development of flexible rechargeable batteries with high-power density and high energy density over a long lifespan[1–3]. Lithium ion batteries (LIBs) have obtained the success of commercialisation owing to the environmentally benign production process, long cycle life, and high battery potential. Nevertheless, they still suffer from the lack of advanced electrodes with high reversible capacities and fast charge–discharge rates for the applications of future flexible electronics[4]. In practical LIBs, graphitic carbon materials are used as anode material, which has been the main constrained factor with unsatisfied theoretical capacity ($\sim 372$ mAh g$^{-1}$) for LiC$_6$ and inferior rate performance. Therefore, it is highly desirable to replace graphitic carbons by high-capacity electrode material, such as alloys (Sn, Si and Ge) and metal oxides (Co$_3$O$_4$, SnO$_2$ and GeO$_2$), owing to their higher theoretical capacities ($>1,000$ mAh g$^{-1}$). Till now, many different strategies have been developed for the preparation of such high-capacity electrodes. Despite these promising characteristics, the major bottlenecks limiting the practical applications of the high-capacity electrode material is largely related to its poor structure stability because of the huge volume change and resultant capacity fade[5–10]. On the other hand, conventional battery structure does not show a good flexibility owing to the weak bonding between the electrode material and the Al or Cu foil (that is, current collector), which tends to detach after repeated bending. In addition, the kinetic limitation of lithium ion diffusion through an electrode material on the current collector causes a large electron and ion transport resistance and, as a consequence, loss of fast charge–discharge rate capability[11,12].

To tackle above challenges, numerous research efforts have been directed to three-dimensional (3D) interconnected graphene network (such as aerogels, hydrogels and foams) as current collector or matrix via hydrothermal, layer-by-layer assembly and chemical vapour deposition methods in recent years[13–17]. Chemical vapor deposition (CVD) is a promising method to synthesize graphene with higher conductivity than other chemical synthesis, which would facilitate the fast electron and ion transport in advanced energy storage system. In this respect, Chen and his co-workers[17] first developed a CVD method for the synthesis of 3D interconnected porous graphene foams by using Ni foams as template and methane (CH$_4$) as carbon source in 2011. Inspired by this work, numerous researches have performed to prepare 3D interconnected graphene-based network structures for advanced electrochemical energy storage applications in the field of supercapacitors, LIBs and lithium-oxygen batteries owing to their adaptive properties such as excellent electrical conductivity, large surface area and good mechanical strength[18–20]. Moreover, the ultra-lightweight and excellent flexibility of 3D interconnected graphene network structures have been identified to be ideal as current collector or matrix for high-capacity electrode materials[21–23]. 3D graphene-based electrode architecture showed good electrochemical performance due to the 3D interconnected porous network structural design provides lager surface area for supporting active materials that facilitates rapid electron and ion diffusion. However, this strategy does not provide enough void space to alleviate the huge volume changes during lithium alloying and leaching, result in the pulverisation, exfoliation and aggregation of electrode materials. Very recently, yolk-shell nanoarchitecture, a new type of strategy, has attracted much attention in many fields, particularly in the areas of energy storage[24–28]. The significant advantage of the yolk-shell nanoarchitecture in enhancing electrochemical property lies in the internal void space, which would be expand/contract freely during lithium alloying and leaching without damaging the outer shell. Therefore, it is envisaged that the incorporation of the yolk-shell structure and 3D interconnected graphene-based electrode architecture is a facile and general approach to enhance the flexibility and electrochemical property of high-capacity electrode.

Among the high-capacity anodes, germanium (Ge) is a potential candidate to replace the commercial graphite anode for LIBs because of its high volumetric and gravimetric capacities (7,360 mAh cm$^{-3}$ and 1,626 mAh g$^{-1}$), low electrochemical potential of lithiation/delithiation ($<0.5$ V versus Li$^+$/Li), excellent lithium diffusivity (400 times faster than in Si) and higher intrinsic electrical conductivity compared with Si[29–31]. Furthermore, it is noteworthy that heteroatom doping (for example, N) of carbon nanomaterials (such as graphene) have been confirmed to greatly improve the electrical conductivity, where the surface hydrophilicity of carbon nanomaterials to favour the charge transfer and electrode–electrolyte interactions[32].

Herein, we show a facile and general approach to a 3D interconnected porous nitrogen-doped graphene foam (NGF) with encapsulated Ge quantum dot@nitrogen-doped graphene yolk-shell nanoarchitecture (Ge-QD@ NG/NGF) for high specific reversible capacity (1,220 mAh g$^{-1}$), ultra-high rate capability (over 800 mAh g$^{-1}$ at 40 C) and long cycling capability (over 98% specific reversible capacity retention from the second to 1,000 cycles) in the form of flexible LIBs. The unique advantage of 3D Ge-QD@NG/NGF yolk-shell nanoarchitecture not only provide internal void space to alleviate the huge volumetric expansion of Ge during lithiation, but also provide numerous open channels for the easy access of electrolyte, good flexibility and great retain in the high electrical conductivity of whole electrode, thus facilitating the fast lithium ion and electron diffusion. The unique 3D interconnected graphene-based electrode structure reported in this work can have significant indications in developing a new generation high-performance flexible LIBs with great promises to promote the real-life applications in flexible energy-storage devices.

## Results

**Materials synthesis and characterisation.** As shown in Fig. 1, the preparation of the Ge-QD@NG/NGF yolk-shell nanoarchitecture is a relatively simple process. First, we selected 3D interconnected porous Ni foam as the template to grow N-doped graphene. Specifically, carbon and nitrogen were deposited into Ni foam by decomposing pyridine at 900 °C for 5 min in flowing Ar (90%)/H$_2$ (10%) gas mixture. N-doped graphene were well-distributed deposited on the 3D interconnected porous Ni foam by CVD. The homogeneous nonaggregated GeO$_2$ nanoparticles were then deposited by using GeCl$_4$ and loaded in 3D interconnected porous N-doped graphene network with Ni foam (GeO$_2$/NG-NF) matrix. The hydrothermal process of the solutions above was a comprehensive way for the obtaining of GeO$_2$ nanoparticles under mild conditions. The GeO$_2$/NG-NF was then coated with Ni thin layer using electroplating deposition. The Ni thin layer serves as not only the catalyst for N-doped graphene growth, but also the sacrificial coating layer for providing internal void space. In this process, we uniform coated GeO$_2$ nanoparticles with Ni-coating layer (GeO$_2$@Ni) core-shell structure. And thickness of the Ni-coating layer may be easily tuned for the appropriate void space by changing the electroplating deposition parameters. It is noted that the key design of the yolk-shell nanoarchitecture in improving electrochemical performance lies in the appropriate void space, which would be expand/contract freely upon lithium alloying and leaching without damaging the

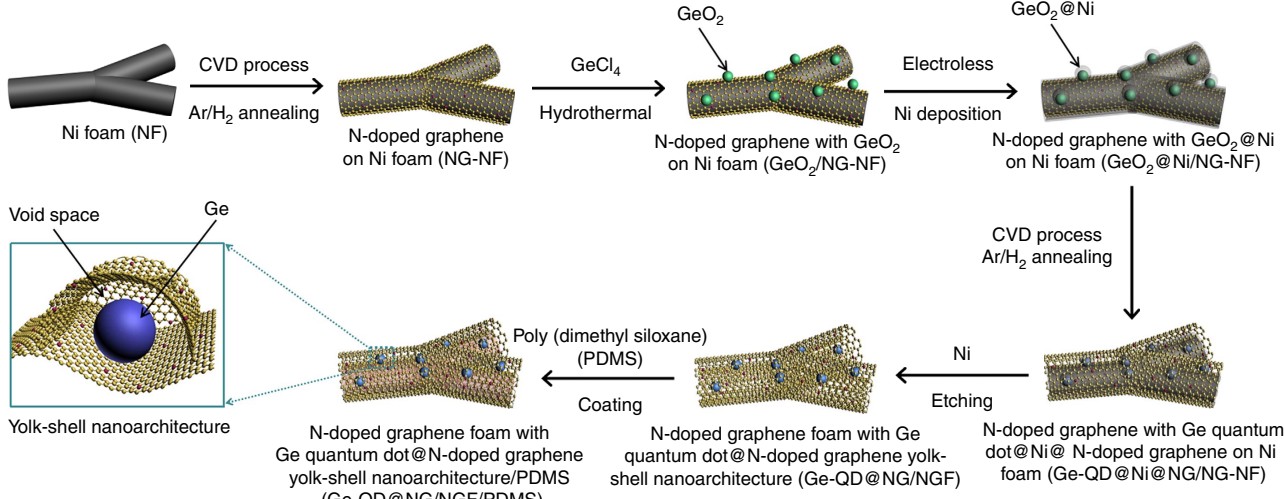

**Figure 1 | Preparation process of Ge-QD@NG/NGF/PDMS structures.** A schematic illustration of the preparation of Ge-QD@NG/NGF/PDMS yolk-shell nanoarchitecture.

outer shell. Thereafter, we catalysed the GeO$_2$@Ni/NG-NF nanoarchitecture for conformal N-doped graphene growth by CVD at 650 °C for 2 min with the same mixed atmosphere as the first step. After that, Ge was generated through the reduction of GeO$_2$ and thermally annealed at 650 °C for 6 h in flowing Ar (90%)/H$_2$ (10%) gas mixture without pyridine. After that acid etching was used to remove the Ni foam and sacrificial layer to obtain the Ge-QD@NG/NGF yolk-shell nanoarchitecture. As a final step, in order to test the flexibility and electrochemical property of the nanoarchitecture, the thin layer of poly (dimethyl siloxane) (PDMS) was uniform coated on the surface of the Ge-QD@NG/NGF yolk-shell nanoarchitecture (Ge-QD@NG/NGF/PDMS). It is noteworthy that PDMS gel was prepared by intensively stirring base/curing agents (Sylgard 184; Dow Corning), then by degassing in vacuum for 1 h and thermally curing for 6 h under 80 °C. As shown in Fig. 2a,b, the Ge-QD@NG/ NGF/PDMS yolk-shell nanoarchitecture delivers excellent flexibility, and can be randomly bent without damage.

The scanning electron microscope (SEM) images of the Ge-QD@NG/NGF yolk-shell nanoarchitecture exhibits the 3D interconnected porous graphene network architecture (Fig. 2c). Only 3D interconnected porous graphene network can be observed in the SEM analysis, whereas the Ge-QD@NG are too small to be seen in the Fig. 2c. The chemical mappings confirmed that Ge-QD@NG was uniformly distributed on the 3D nitrogen-doped foam, where N atoms have been doped in the graphene by decomposing pyridine as the nitrogen and carbon sources (Fig. 2d)[33]. In order to directly observe the encapsulated Ge-QD@NG nanoarchitecture, the uniform morphologies were investigated by the transmission electron microscopy (TEM) (Fig. 2e). The Ge-QD@NG nanoarchitectures were seen to be uniform dispersed on 3D NGF with size range of 3–7 nm (Fig. 2a). As indicating in the images, Ni as the shell effectively inhibiting the further growth of GeO$_2$ nanoparticles as core in the process of thermal reduction. To further examine the unique structure of the Ge-QD@NG yolk-shell nanoarchitecture, the uniform morphologies were investigated by the high-resolution TEM. As shown in Fig. 2f, Ge-QD is encapsulated by a self-supporting N-doped graphene outer shell, which may be limiting solid/electrolyte interphase (SEI) formation to the outer shell surface. The Ge core is closely attached to one side of the 3D NGF matrix, leaving a ~2.2 nm internal void space. The direct contact of the Ge-QD core with the electrically and lithium ion conductive NGF enables the core more accessible to both

charge carriers. The hollow interior of the yolk-shell nano-architecture is clearly revealed under TEM observation (Fig. 2f). Inside the structure, Ge-QD (~5 nm in diameter) are encapsulated by the N-doped graphene shell with well-defined void space. The measured interplanar distance of ~0.32 nm corresponds well with the (111) planes of the diamond cubic phase of Ge. The selected area electronic diffraction pattern (Fig. 2g) corresponding to Ge-QD gives a set of diffraction rings, which can be clearly assigned to the diffractions of the (111), (220) and (311) planes, respectively.

X-ray diffraction (XRD) is conducted to investigate the crystallographic phases of the Ge-QD@NG/NGF yolk-shell nanoarchitecture. The XRD pattern of the nanoarchitecture in Fig. 2h shows peaks well-corresponding to the diamond cubic phase of Ge (JCPDS No. 04–0545)[34]. The diffraction peaks for N-doped graphene are absent, which is due to the fact that they are likely to be eclipsed by the Ge (111) peak, in the range of 25–28°. Although the presence of N-doped graphene in the sample was not evidenced by the XRD data, its existence was determined by Raman scattering (Fig. 2i). The appearance of the strong peak at 1588 cm$^{-1}$, weak peaks at 981 cm$^{-1}$ and 927 cm$^{-1}$, demonstrating the existence of N-doped graphene in the obtained nanoarchitecture[35]. The very sharp peak observed at 296 cm$^{-1}$ was assigned to crystalline Ge[36]. In addition, the mass ratio of Ge in the Ge-QD@NG/NGF yolk-shell nanoarchitecture was measured by thermal gravimetric analysis (Supplementary Fig. 1). The mass of the Ge component increased slightly at 700 °C owing to the oxidation of Ge; the NGF had almost completely burned away at 700 °C (that is, it lost 99.7% of its weight). This thermal gravimetric analysis result allows the mass ratio of Ge to N-doped graphene to be calculated by the mass difference of Ge, NGF and Ge-QD@ NG/NGF yolk-shell nanoarchitecture, thus 73.76 wt% of Ge and 26.24 wt% of N-doped graphene (Supplementary Table 1 and Supplementary Note 1). For comparison, the pure Ge nanoparticle (Ge) and 3D interconnected porous nitrogen-doped graphene foam with encapsulated pure Ge nanoparticles (Ge/NGF) architecture were also synthesized by using the same experimental condition without Ni-coating layer (Supplementary Figs 2a and 3). It is clear from the TEM image that the Ge nanoparticle without Ni-coating layer has been further growth under the thermal reduction process with an average diameter of ~20 nm (see Supplementary Figs 2b and 4). It is noted that the presence of the Ni-coating layer can restrict the further growth of Ge nanoparticles under the

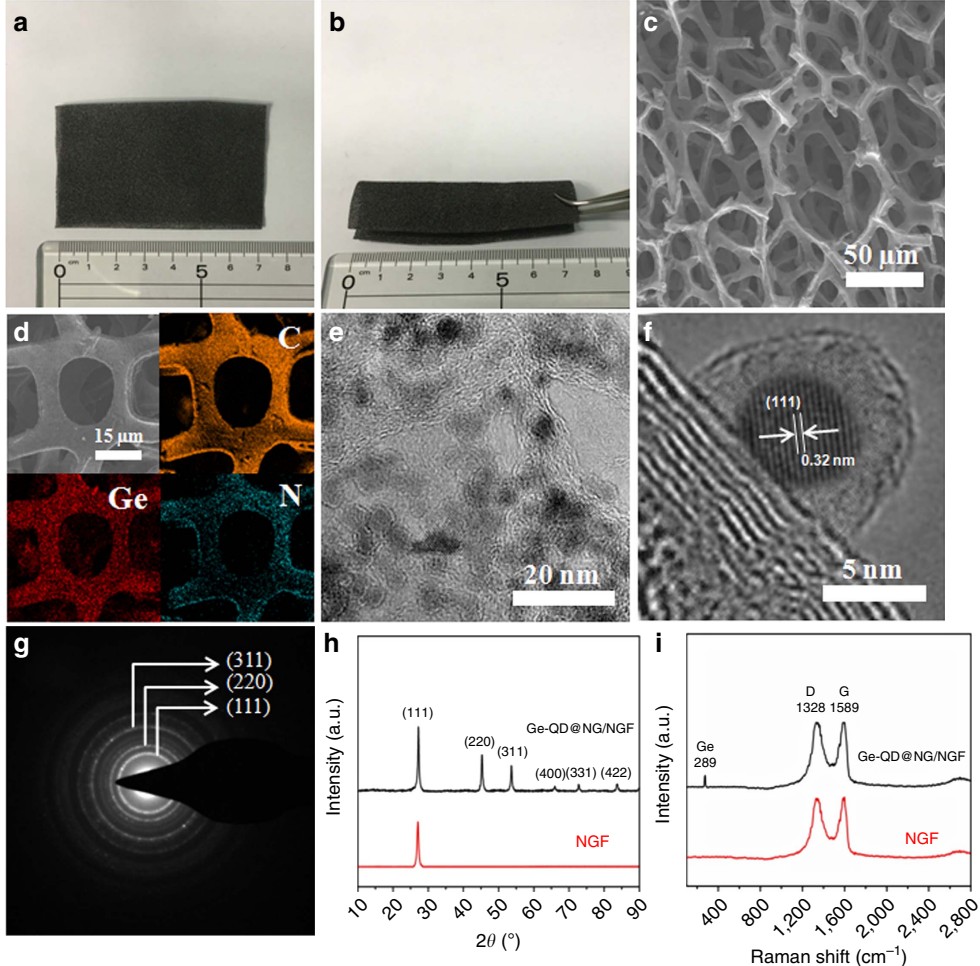

**Figure 2 | Physical characterisation of the Ge-QD@NG/NGF.** (**a,b**) Photographs of a flexible Ge-QD@NG/NGF yolk-shell electrode (7 × 4 cm). (**c**) SEM image of the Ge-QD@NG/NGF yolk-shell nanoarchitecture. (**d**) EDS elemental maps of Ge, C and N, respectively. (**e**) TEM image of the Ge-QD@NG/NGF yolk-shell nanoarchitecture. (**f**) High-resolution TEM of the Ge-QD@NG/NGF yolk-shell nanoarchitecture. (**g**) The electronic diffraction pattern corresponding to the Ge-QD. (**h**) XRD patterns and (**i**) Raman spectra of the Ge-QD@NG/NGF yolk-shell nanoarchitecture and NGF.

thermal reduction process. In addition, a total specific surface area of 392.8 m$^2$ g$^{-1}$ was obtained by the Brunauer-Emmett-Teller method (see Supplementary Fig. 5). Such high surface area provides more surface-active sites and makes the diffusion of the liquid electrolyte into the electrodes more easily, leading to an enhancement of the electrochemical performance.

**Electrochemical performance.** The rational design and fabrication of the Ge-QD@NG/NGF yolk-shell nanoarchitecture for a stabilized electrode is evident from the outstanding electrochemical performance (see Fig. 3). In order to test the flexibility and electrochemical property of the nanoarchitecture, we weighed the same quality of PDMS by spin coating to uniform coat on the surface of the Ge-QD@NG/NGF yolk-shell nanoarchitecture (Ge-QD@NG/NGF/PDMS) and Ge/NGF nanoarchitecture (Ge/NGF/PDMS). The electrochemical property of Ge-QD@NG/NGF/PDMS yolk-shell electrode was first evaluated by galvanostatic charge–discharge measurements in the 0.01–1.5 V voltage window (Fig. 3a). The voltage profile with different flat plateaus because of the redox reactions associated with the process of lithiation/delithiation may be seen by the first charge and discharge curves. In the first cycle, the Ge-QD@NG/NGF/PDMS yolk-shell nanoarchitecture can achieve high columbic efficiency at the first cycle in contrast to a large initial irreversible capacity

loss observed for the pure Ge and Ge/NGF electrodes. The discharge and charge capacities of the Ge-QD@NG/NGF yolk-shell nanoarchitecture are 1,597 mAh g$^{-1}$ and 1,220 mAh g$^{-1}$, respectively, corresponding to a initial columbic efficiency of ∼76.39%, which is higher than that Ge and Ge/NGF electrodes. The irreversible capacity ratio can be assigned to the decomposition of electrolyte, forming a SEI on the surface of the electrode, and to the irreversible insertion of lithium ions into the Ge[37,38]. Figures 3c,d show a series of Raman spectra of the Ge-QD@NG/NGF/ PDMS yolk-shell nanoarchitecture in the working battery during lithiation. Figure 3c shows the schematic of the electrochemical cell. The cells were assembled using a Ge-QD@NG/NGF/PDMS yolk-shell nanoarchitecture electrode, lithium foil was used as the counter electrode, and a porous celgard separator soaked in electrolyte. The first Raman spectrum was collected before the lithiation began, in which the first-order Raman peak occurred at 296 cm$^{-1}$ was assigned to crystalline Ge (Fig. 3d). The other Raman peaks were from either the cell packaging material (such as PDMS) or the electrolyte. The intensity of Ge Raman peak maintains sharp in the first 5 h, followed by the intensity declined during the process of further lithiation. The electrochemical process of Ge anodes can be determined by the change of Raman peak intensity. At beginning stage, Ge nanoparticles remains unreacted with a steady Raman peak intensity of Ge nanoparticle[39]. When the lithiation of

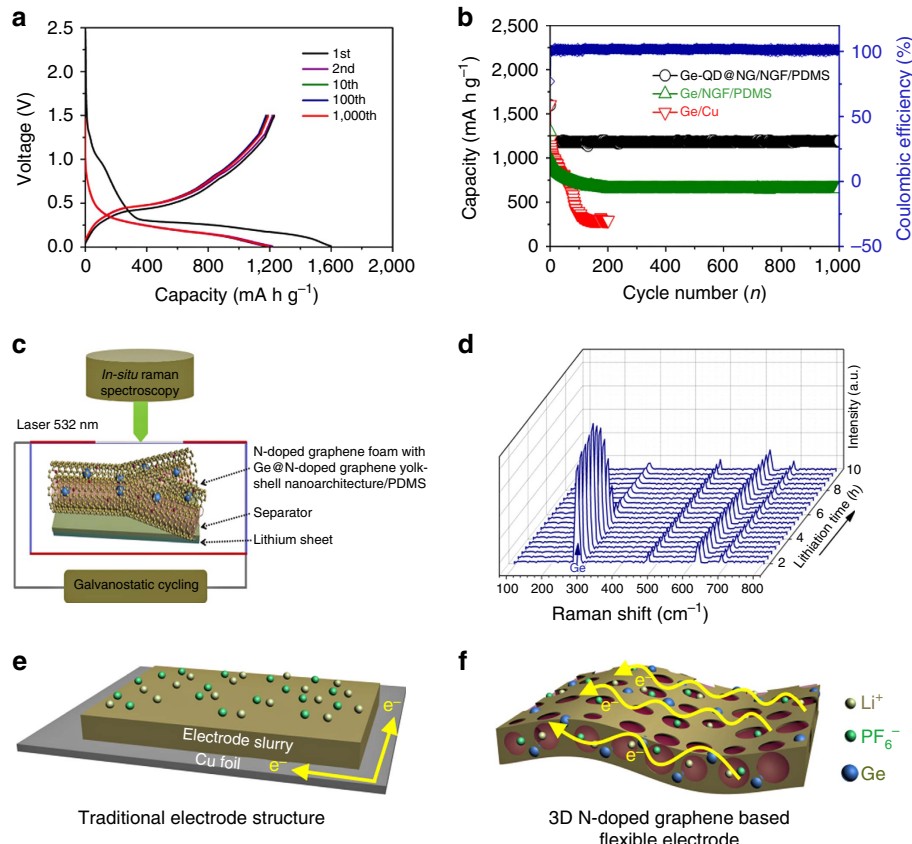

**Figure 3 | Electrochemical performance of the Ge-QD@NG/NGF/PDMS.** (**a**) Galvanostatic charge–discharge profiles in the 0.01–1.5 V window (versus Li/Li$^+$) for the 1st, 2nd, 10th, 100th and 1,000th cycles at 1 C. (**b**) Cycling performance (discharge) and coulombic efficiency of the Ge-QD@NG/NGF/PDMS yolk-shell electrode, Ge/NGF/PDMS and Ge/Cu electrodes at 1 C for 1,000 cycles. (**c**) A schematic diagram of the 'transparent' half cell for *in situ* micro-Raman measurement. (**d**) Selected Raman spectra of the half cell during galvanostatic lithiation of the Ge-QD@NG/NGF/PDMS yolk-shell nanoarchitecture at a rate of C/10. A laser power of 2.5 mW and a collection time of 30 s were used for each spectrum, for each acquisition 10 spectra were accumulated. (**e,f**) Comparison between electrode design in which Ge is coated on an Cu foil and a NGF-based flexible electrode.

Ge electrode starts after 5 h, the surface layer of Ge nanoparticle is converted into a-Li$_x$Ge, which exhibits better conductivity than Ge. The good conductivity of a-Li$_x$Ge leads to a small optical skin depth (penetration depth of the laser), which explicates the reduction of Ge Raman peak intensity. Simultaneously, the intensities of the Raman peaks of the cell packaging material or the electrolyte keep almost constant, demonstrating that the change of Ge Raman peak intensity is only associated with the electrochemical process of the lithiation.

As control experiments, the Ge/NGF architecture were also synthesized and tested for LIBs (Supplementary Figs 2 and 3). As shown in Fig. 3b, a fast capacity fading observed in the pure Ge electrode obviously. The large size of Ge nanoparticles has leaded to poor strain relaxation. And the differential cracking to occur in the nanoparticles was mainly due to the uneven distribution of the induced stress. The situation has been partially strengthened in the case of the Ge/NGF electrodes, which provide more space to accommodate and load electrode material, and act as an interconnected conductive network. Figure 3e,f illustrate the structural differences between the Ge/NGF/PDMS flexible electrode and the conventional electrode with Ge coated on Cu current collector. The current collector has a significant role in loading electrode materials and offering an interconnected conductive path way. Unlike the conventional Cu current collector, the NGF-based current collector with a continuous porous network is conducive to offer better electrical and electrolyte contact[12]. However, on one hand, the large size of

Ge nanoparticles in the Ge/NGF/PDMS electrode has leaded to poor strain relaxation. On the other hand, the Ge/NGF/PDMS electrode does not provide internal void space to alleviate the huge volume changes of Ge during lithium alloying and leaching, which resulted in a gradual capacity decline in the first 200 circles. Remarkably, the Ge-QD@NG/NGF/PDMS yolk-shell electrode has excellent cycle stability (Fig. 3b). There was no obvious change from the 5th to the 1,000th cycle in the shape of the profile, exhibiting stable electrochemical performance of Ge-QD@NG/NGF/PDMS yolk-shell electrode. It is noted that the Ge-QD@NG/NGF/PDMS yolk-shell electrode is distinctly superior to the corresponding Ge/Cu and Ge/NGF/PDMS electrodes, which has an average coulombic efficiency reaches ~99.7% during the 1,000 cycles. Furthermore, reversible capacities ~996, 901 and 792 mA h g$^{-1}$ could be reached at higher charge–discharge rate of 10, 20 and 40 C after 200 cycles (Supplementary Fig. 6). In terms of the mechanical stability of the Ge-QD@NG/NGF/PDMS yolk-shell electrode, the maximal stress that they may sustain reached a plateau of 0.78 M Pa for NGF/PDMS and increased slightly after the deposition of Ge-QD@NG (Fig. 4a,b). We also investigated the effect of bending test on the electrochemistry property of the Ge-QD@NG/NGF/PDMS yolk-shell electrode. Compared with that of the original flat state, only a negligible over-potential and a <2% specific reversible capacity reduction can be observed under bend state (bending angle of 90°) (Fig. 4c). Moreover, the Ge-QD@NG/NGF yolk-shell electrode exhibited outstanding

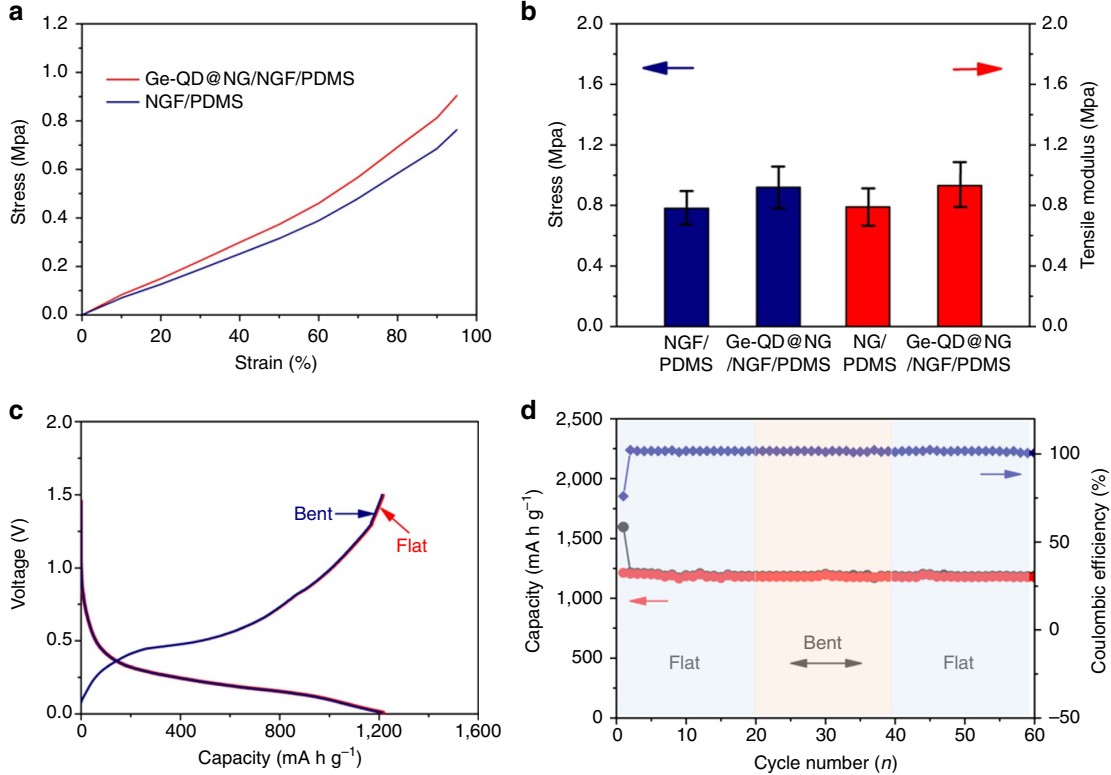

**Figure 4 | Flexible testing of the Ge-QD@NG/NGF/PDMS.** (**a**) Typical stress–strain curves of free-standing 3D electrode structure with and without Ge-QD@NG. (**b**) Tensile strength and modulus of free-standing 3D electrode structure with and without Ge-QD@NG. Date are presented as mean ± s.e.m., and error bars show s.e.m. (**c**) Galvanostatic charging/discharging curves of the battery. Red and blue lines represent the as-fabricated flat battery after 20 cycles and the bent battery after 20 cycles under repeatedly bending (bending angle of 90°), respectively. (**d**) Cyclic performance of the battery under flat and bent states. Red and grey lines represent the charging and discharging performance at charge–discharge rate of 1 C.

cycle stability under flat and bent states. In particular, the Ge-QD@NG/NGF/PDMS yolk-shell electrode exhibited a reversible specific capacity retention of ~96.8% from the second to first 20 cycles under a flat state, and ~95.2% after another 20 cycles under a bent state at charge–discharge rate of 1 C (Fig. 4d). The unique structure of Ge-QD@NG/NGF/PDMS yolk-shell electrode is more stable and flexible, which can effectively buffer the strain from the huge volume expansion/contraction during the lithiation/ delithiation process.

The rate capability of the Ge-QD@NG/NGF/PDMS yolk-shell electrode was investigated at various charge–discharge rates for each of the 10 cycles, as shown in Fig. 5a. The reversible capacity of Ge-QD@NG/NGF/PDMS yolk-shell electrode was 1,194, 1,126, 1,072, 1,001, 905 and 801 mAh g$^{-1}$ at charge–discharge rates of 1, 2, 5, 10, 20 and 40 C, respectively. Subsequently, a rebound in specific capacity with a reduced slightly can be seen for the Ge-QD@NG/NGF/PDMS yolk-shell electrode. The charge–discharge rate was brought back to an initial rate of 1 C at which the specific capacity was 1,185 mAh g$^{-1}$ and thus nearly fully recovered. After undergoing high charge–discharge rate, the specific capacity rapidly returned to the initial value, demonstrating the excellent rate performance of the electrode. In order to understand the excellent rate capability, the Nyquist plots of Ge/Cu, Ge/NGF/PDMS and Ge-QD@NG/NGF/PDMS yolk-shell electrodes (Supplementary Fig. 7). Apparently, the Ge-QD@NG/NGF/PDMS yolk-shell electrode displays a much lower resistance than the Ge/Cu and Ge/NGF/PDMS electrode. Significantly, the 3D interconnected porous NGF structural design is proved to be useful in minimising electron and ion transport resistance. Moreover, the N-doped graphene outershell

acts as a channel for lithium ion to Ge during lithium alloying and leaching, which is indicated by the high diffusion coefficient of N-doped graphene compared with Ge. Therefore, the incorporation of the yolk-shell structure and 3D interconnected porous electrode architectures have enable a higher diffusion rate of lithium ion, improved conductivity and enhanced rate capability, all of which are crucial for practical use in LIBs.

To better understand the excellent cycle stability, electrochemical impedance spectroscopy (EIS) analysis of the Ge-QD@NG/NGF/PDMS yolk-shell electrode was conducted after the 1st, 2nd, 10th, 100th and 1,000th cycles at a charge–discharge rate of 1 C (Fig. 5b). No obvious resistance increase was occurred after cycling, demonstrating the good structure stability of the electrode. As determined by the FESEM and TEM images of Ge-QD@NG/NGF/PDMS yolk-shell electrode in lithium intercalation state after 1,000 cycles (Fig. 5c), the uniform morphology and yolk-shell structure of the Ge-QD@NG/NGF/PDMS electrode is well retained (Fig. 5d). TEM image of a lithiated yolk-shell nanoarchitecture shows a distinct N-doped graphene coating (Fig. 5e), indicating the Ge-QD core not only may expand/contract freely during lithium alloying and leaching without damaging the N-doped graphene outer shell. This result clearly demonstrates that the incorporation of the yolk-shell structure and 3D interconnected porous graphene-based electrode architectures can accommodate the significant volume expansion/contraction in Li-Ge alloying and de-alloying reactions during the charge–discharge processes, which is the key factor for a high reversible capacity, excellent cycle stability and ultra-high rate capability electrode. As shown in Fig. 5f, the excellent cycle stability and ultra-high rate capability of Ge-QD@NG/NGF

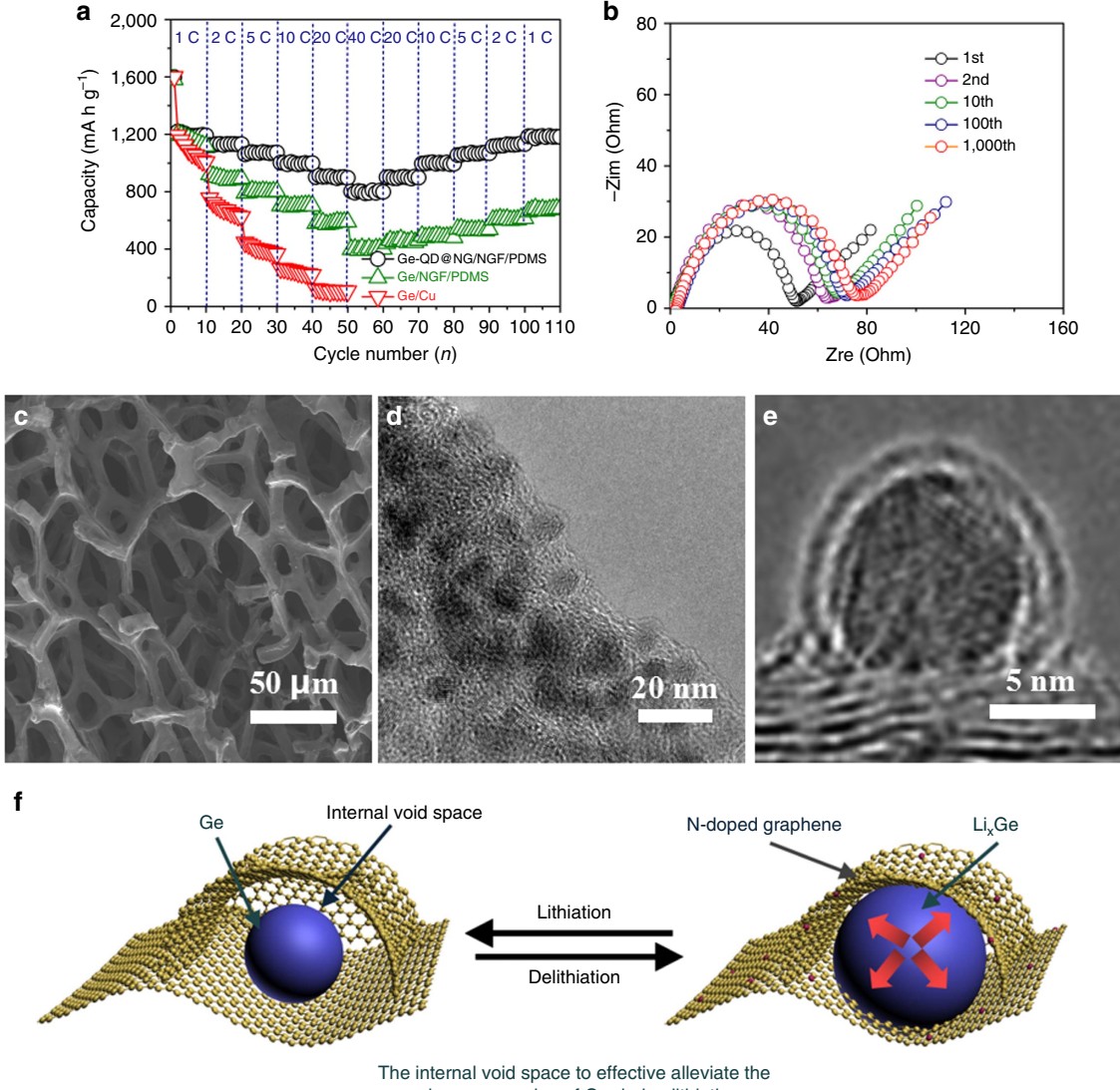

**Figure 5 | Rate performance and morphological changes of the Ge-QD@NG/NGF/PDMS. (a)** Rate performance of the Ge-QD@NG/NGF/PDMS yolk-shell, Ge/NGF/PDMS and Ge/Cu electrodes at different current densities. **(b)** Nyquist plots of the Ge-QD@NG/NGF/PDMS yolk-shell electrode after the 1st, 2nd, 10th, 100th and 1,000th cycles at a current density of 1 C. **(c–e)** SEM **(c)** and TEM **(d,e)** images of Ge-QD@NG/NGF/PDMS yolk-shell electrode in lithium intercalation state after 1,000 cycles at a current density of 1 C. **(f)** Schematic drawing of the charge/discharge processes of the Ge-QD@NG/NGF/PDMS yolk-shell electrode.

yolk-shell nanoarchitecture may be attributed to the following mechanism: (i) The electrode materials, Ge, has been homogeneously dispersed with a size down to several nanometres. In such a manner, the compact contact between electrode materials and current collector can be achieved, which is favourable for electrode materials activation processes. It is also believed that quantum-size confinement effect for lithium storage can occur in surface region of electrode materials and therefore improve the specific capacity[40,41]. (ii) The design of yolk-shell structure not only effectively alleviate the huge volume expansion/ contraction during charge–discharge process, but also make the SEI to form mainly on the N-doped graphene outer shell, which has seal any possible defects and prevent electrolyte infiltration through the conformal outer shell[24–28]. (iii) The entire 3D interconnected porous electrode system was formed with a well-defined porosity, maximized surface area, and enlarged lattice spacing between graphene layers, coupled with the N-doping-induced defects. Such structure is beneficial for the synergistic effects, which

facilitate the fast lithium ion and electron diffusion, improve the storage of lithium ions and accommodate the effect of huge volume change during the lithiation/delithiation process[42–44].

## Discussion

In summary, Ge quantum dot based anode materials for high performance LIBs have been demonstrated by design of a 3D interconnected porous nitrogen-doped graphene foam with encapsulated Ge quantum dot@nitrogen-doped graphene yolk-shell nanoarchitecture. The successful creation of internal void space afforded a better alleviating of the volume expansion during lithiation. Futhermore, the N-doped graphene outer shell has played a major role in minimising the pulverisation, exfoliation and aggregation of Ge. Compared with pure Ge and Ge/NGF/ PDMS electrodes, the Ge-QD@NG/NGF/ PDMS exhibited high specific reversible capacity (1,220 mAh g$^{-1}$), long cycling capability (over 96% reversible capacity retention from the second

### Table 1 | Experimental parameters.

| Composition | Content per g l$^{-1}$ |
|---|---|
| Nickel sulfate | 70 |
| Nickel chloride | 35 |
| Boric acid | 35 |
| Code position condition | |
| Temperature | 50 |
| pH | 4.0 |

The composition of electroplating bath and processing parameters.

to 1,000 cycles) and ultra-high rate capability (over 800 mAh g$^{-1}$ at 40 C), which is one of the highest known values (Table 1)[45–54]. Therefore, it is believed that our work opens up new opportunities toward realising the practical applications of 3D interconnected graphene-based high-capacity flexible electrode material, particularly those that suffer from huge volume expansion.

## Methods

**Preparation of Ge-QD@NG/NGF yolk-shell nanoarchitecture.** First, the porous N-doped graphene foam structure was synthesized by a CVD technique using the porous Ni foam as the template with pyridine as the nitrogen and carbon sources for N-doped graphene growth. The porous Ni foam was cut into pieces of 7 × 4 cm and placed in a quartz tube furnace and annealed at 900 °C in flowing Ar (90%)/H$_2$ (10%) gas mixture for 20 min to reduce the surface oxide layer. After that, pyridine was decomposed under the mixed atmosphere of Ar (90%)/H$_2$ (10%) gas for N-doped graphene growth for 5 min under 900 °C. Then, GeCl$_4$ (99.99%) was dissolved in ethanol to form a solution (0.1 mol l$^{-1}$). The obtained N-doped graphene with porous Ni foam was soaked in above solution before being transferred to a 100 ml Teflon-lined autoclave and hydrothermally treated at 100 °C for 10 h. The N-doped graphene with porous Ni foam covered with GeO$_2$ nanoparticles (GeO$_2$/NG-NF) was washed using deionized water and dried under vacuum at 80 °C. After that, GeO$_2$@Ni/NG-NF core-shell nanoarchitecture was prepared using electroplating deposition method. The composition of the Ni bath and the electroplating deposition parameters are shown in Table 1.

A pure Ni plate (99.98 wt.%) was used as anode. GeO$_2$/NG-NF was used as cathode. The electroplating deposition was performed on a type CHI660D electrochemical workstation. The forward (cathode) current density is set to 20 mA cm$^{-2}$. Electroplating deposition total time is 5 min. The resulting GeO$_2$@Ni/NG-NF core-shell nanoarchitecture was washed using ethanol, and then dried for 6 h at 80 °C under vacuum. Afterwards, as-prepared sample was used as the template for growing one more layer of N-doped graphene by decomposing pyridine at 650 °C for 2 min in flowing Ar (90%)/H$_2$ (10%) gas mixture. Subsequently, annealed at 650 °C in flowing Ar (90%)/H$_2$ (10%) gas mixture without pyridine for 6 h to reduce the GeO$_2$ into Ge. Finally, the interconnected porous NGF with encapsulated Ge quantum dot@nitrogen-doped graphene yolk-shell nanoarchitecture (Ge-QD@ NG/NGF) was prepared after etching the Ni foam in 1 M HCl. The content of Ge may be controlled through changing the concentration of GeCl$_4$ in the above solution.

**Preparation of Ge nanoparticle.** GeCl$_4$ (99.99%) was dissolved in ethanol to form a solution (0.1 mol l$^{-1}$) and soaked in above solution before being transferred to a 100 ml Teflon-lined autoclave and hydrothermally treated at 100 °C for 10 h. Then as-synthesized sample was recovered using centrifugation and dried under vacuum chamber overnight further characterisation.

**Preparation of Ge/NGF nanoarchitecture.** First, the porous NGF structure was synthesized by a CVD technique using the porous Ni foam as the template with pyridine as the nitrogen and carbon sources for N-doped graphene growth. The porous Ni foam was cut into pieces of 7 × 4 cm and placed in a quartz tube furnace and annealed at 900 °C in flowing Ar (90%)/H$_2$ (10%) gas mixture for 20 min to reduce the surface oxide layer. After that, pyridine was decomposed under the mixed atmosphere of Ar (90%)/H$_2$ (10%) gas for N-doped graphene growth for 5 min under 900 °C. Then, GeCl$_4$ (99.99%) was dissolved in ethanol to form a solution (0.1 mol l$^{-1}$). The obtained N-doped graphene with porous Ni foam was soaked in above solution before being transferred to a 100 ml Teflon-lined autoclave and hydrothermally treated at 100 °C for 10 h. The GeO$_2$/NGF were kept in a vacuum oven for 10 min under room temperature, and heated at 650 °C with the protection of Ar (90%)/H$_2$ (10%) atmosphere for 6 h. After that, the resulting powder was etched with 1 M HCl solution to remove Ni foam. The final samples were kept in a vacuum chamber to avoid oxidation for further characterisation.

**Material characterisation.** The morphology, composition and crystalline phase of the as-prepared samples were characterized by field-emission scanning electron microscopy (FESEM; S-4800, Hitachi), TEM and high-resolution transmission electron microscopy (JEM-2100). XRD measurements were conducted on a Rigaku D/max-γB diffractometer using Cu Kα radiation at a scan rate of 2° min$^{-1}$. Thermogravimetric analysis was measured using a TG/DTA6200 instrument. The strength of the nanoarchitecture was carried out using a high-precision mechanical testing system (Instron 5500 Rmaterials tester). In situ Raman analysis was measured with a Renishaw in Via micro-Raman system with a 532 nm laser excitation line.

**Electrochemical measurements.** For half cell tests of free-standing Ge/NGF/ PDMS and Ge-QD@NG/NGF/ PDMS yolk-shell nanoarchitecture were fabricated. In particular, the mass loading of electrode materials was calculated at ca.1.8 mg cm$^{-2}$ and assembled into two-electrode CR2025-type coin cells in an Ar-filled glove box for next battery measurements, porous polypropylene film was used as the separator, lithium foil as the counter electrode. 1 M solution of LiPF$_6$ in a volumetric ratio of 1:1:1 mixture of ethylene carbonate and dimethyl carbonate and diethyl) was used as an electrolyte. The charge and discharge performance was tested at different charge–discharge rates under room temperature. It's remarkable that all the electrochemical data were calculated based on the total mass of the composite including graphene and Ge in this work. All tests on the free-standing Ge/NGF/PDMS and Ge-QD@NG/NGF/PDMS yolk-shell nanoarchitecture were conducted without Cu current collector, binder and carbon black. The traditional Ge anode was fabricated by mixing the Ge nanoparticle, carbon black, and poly-vinylidene fluoride at a weight ratio of 8:1:1. The electrode slurry was pasted onto a Cu current collector and dried for 12 h under 120 °C in vacuum. All coin cells were assembled in an Ar-filled glove box. The charge and discharge performance was carried out on the BTS battery test system (NEWARE battery test system (BTS)-610) within a potential range of 0.01–1.5 V (versus Li$^+$/Li) at different charge–discharge rates. EIS measurement was performed on a Parstat 2273 advanced electrochemical system with the frequency range from 100 kHz to 10 mHz at open circuit voltage by applying a 5 mV signal. All of the capacities and C-rate currents in this work were calculated based on Ge anode (1 C corresponding to 1,600 mA g$^{-1}$ for Ge).

**Data availability.** All relevant data supporting the findings of this study are available from the authors on request.

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

## Acknowledgements

This work is supported by Singapore Ministry of Education Academic Research Fund Tier 2 (MOE2015-T2-1-150) and Singapore National Research Foundation Funds. This work is also financially supported by National Natural Science Foundation of China (NSFC 21376001), Jiangsu Province Cultivation base for State Key Laboratory of Photovoltaic Science and Technology (201508).

## Author contributions

R.M., H.Y.Y. and K.S. conceived the idea. R.M. carried out material synthesis and electrochemical tests. R.M., H.Y.Y. and R.N. wrote the paper. All the authors discussed the results and commented on the manuscript.

## Additional information

**Competing financial interests:** The authors declare no competing financial interests.

