## [Peer Review File · Nature Communications]

Reviewers' comments:

Reviewer #1 (Remarks to the Author):

This article describes the development of a 3D interconnected porous nitrogen-doped graphene foam with encapsulated Ge quantum dot@nitrogen-doped graphene (Ge-QD@NG/NGF) yolk-shell nanoarchitecture for use in flexible Li-ion batteries. It demonstrates a promising performance (especially the long cycling behavior for up to 1000 cycles). Although the yolk-shell architecture for alloy materials for use in Li-ion batteries is not novel ((e.g. highlighted in: Chem. Commun., 2011, 47, 12578-12591), it is the first time that this is demonstrated for Ge, to the best of my knowledge. However, it is not clear in the manuscript how the performance is compared to other Ge nano-architectures in the literature and this should have been definitely included in the discussion part. It could be of the interest for the readers of Nature Communications working with alloy compounds for Li-ion batteries, as well as those working with functional flexible graphene structures.

The methodology followed is reasonable, including synthesis, structural characterisation and electrochemical performance parts. However, it is quite disappointing that it was not thoroughly proof-read before submitting to Nature Communications, as this distracts somewhat from the science. For example, the discussion about the properties (pg. 8-13) is difficult to follow, since it is not presented in a logical order. Starting with Figure 3(a), continues to 4(a) and 4(b), then goes back to 3(c), 3(e) and 3(f), followed by a jump to Figure 5 (a,b,c,d in an order), back to Figure 3(d), 3(b), then 4(c), 4(d), 4(e) and finishes with 4(f). Either the figures need to change to follow the order of the text, or the text needs to change to reflect the order of the figures. There are also missing parts in the figures, especially in Figure 3, which demonstrates the electrochemical performance of the material and is actually the strong point of this communication, according to the abstract, and hence it should be very well presented. For instance, in Figure 3(c), the y² axis is presumably columbic efficiency (%) as mentioned in the caption, however this is missing on the figure and there is no connection to the blue line, which is probably corresponding to the coulombic efficiency. It should also be mentioned that Figure S6, which is discussed in pg. 12 does not exist! Moreover, the acronyms are not defined clearly and this can be confusing to a reader who is not familiar with the terminology. For example, the acronym for the polymer used (PDMS, presumably polydimethylsiloxane) is not given at all in the text. Also, it would help if the acronyms, which are used in the text (pg. 5-6) at each step of the synthesis procedure, were indicated within the relevant schematic in Figure 1.

The structural characterisation of the Ge-QD@NG/NGF nano-architecture, is well supported by XRD, TEM, and Raman, in Figure 2. However, it is not clear how the samples for comparison (Ge nanoparticle and Ge/NGF) were made. This could possibly be added in Figure 1 and introduce the acronyms used in the rest of the text. The characterisation for Ge nanoparticle (Ge) sample is shown in Figures S1 and S2 and is convincing, although a comment about the estimated size of Ge nanoparticles, as shown in Figure S2, would be necessary. The characterisation for Ge/NGF (Figure S3, which is not discussed at all at the text) is quite poor. TEM would help to indicate the difference with the Ge-QD@NG/NGF nano-architecture. Both for the Ge and Ge/NGF samples, the axis of XRD (Figures S1 and S3) should extent to 90 °2theta degrees for consistency with the XRD pattern for the Ge-QD@NG/NGF (Figure 2h).

The Ge content in Ge-QD@NG/NGF nano-architecture was estimated by thermogravimetric analysis (TGA, Figure S4), by using information for the Ge and Ge/NGF samples and the analysis is shown in Table S1. It should be mentioned that it is not clear how the equation used is derived and whether the terms WNG, WGe, WGe-QD@NG/NGF, XGe correspond to the tabs in the table. Also, the PDMS polymer would be expected to burn off at some temperature for the Ge-QD@NG/NGF sample; this is not mentioned at all (if not, a reference should be given). Finally, the discussion about the TGA results (pg.8) refers to 700 °C, however it is not clear as to where this data point comes from, since in the analysis in Table S1 the maximum temperature is 675°C.

The electrochemical performance is shown well, although some rearrangement of the figures and/or text would be necessary, as aforementioned. However, there is no evidence that the lithiated phase is $\text{Li}_{4.4}\text{Ge}$ as shown in the schematic in Figure 4(f) and this is not discussed in the text. Figure 4(e) is an important figure, aiming to demonstrate the Li intercalation in the structure and the retention of the yolk-shell architecture after 1000 cycles, by TEM. It would be good to do in-situ TEM to follow the insertion of Li into Ge-QD@NG/NGF, if possible. Figure 4 (b) shows the results of in-situ Raman during lithiation of Ge-QD@NG/NGF, however is not discussed much in the text and no references are given to support the argument that Li is actually incorporated in the structure. Finally, not much information is given for the PDMS coating, as this is described in the synthesis part (page 6) but not at the experimental section (pg.16). Its role (possibly linked with the observed flexibility?), is not discussed in the text.

At this point, I am unsure whether to support this article for publication in Nature Communications. The demonstrated promising performance needs to be compared and discussed in the general context of Ge nano-architectures and other metal containing yolk-shell nano-architectures existing in the literature for Li-ion batteries. It also needs serious re-writing in a reader-friendly manner and some scientific evidence for the claims presented.

Reviewer #2 (Remarks to the Author):

This manuscript introduced superior rate capability and cycling stability of flexible lithium ion battery anode prepared from 3D interconnected porous nitrogen-doped graphene foam with encapsulated Ge quantum dot@nitrogen-doped graphene yolk-shell nanoarchitecture. By the rational design of novel electrode nanoarchitecture, structure agglomeration and volume change are greatly alleviated, and faster electronic and ionic paths are provided. The idea was cleverly conceived and experiments were carefully carried out. This work would make big impacts to the field of high capacity anode materials with large volume change for flexible lithium ion batteries. The paper is ready to be published after addressing some minor concerns.

1. The authors need to clarify what weight basis of the specific capacity is calculated on. Is it based on the total weight of the nanoarchitecture or based on the Ge quantum dot only? Without this information, it is hard to compare the performance with other reported one.

2. Can the author give the specific surface area of the as-prepared Ge-QD@NG/NGF yolk-shell nanoarchitecture?

3. The authors need to provide the information on the total weight of the materials when fabricated as electrodes. Too little material will cause much error when calculating the specific capacity.

4. The authors should pay particular attention to English grammar and sentence structure to make the description be clear to the readers.

Reviewers' comments:

Reviewer #1 (Remarks to the Author):

Comment 1: This article describes the development of a 3D interconnected porous nitrogen-doped graphene foam with encapsulated Ge quantum dot@nitrogen-doped graphene (Ge-QD@NG/NGF) yolk-shell nanoarchitecture for use in flexible Li-ion batteries. It demonstrates a promising performance (especially the long cycling behavior for up to 1000 cycles). Although the yolk-shell architecture for alloy materials for use in Li-ion batteries is not novel ((e.g. highlighted in: Chem. Commun., 2011, 47, 12578-12591), it is the first time that this is demonstrated for Ge, to the best of my knowledge. However, it is not clear in the manuscript how the performance is compared to other Ge nano-architectures in the literature and this should have been definitely included in the discussion part. It could be of the interest for the readers of Nature Communications working with alloy compounds for Li-ion batteries, as well as those working with functional flexible graphene structures.

Response/corrections: We sincerely appreciate the Reviewer's insightful comments, which greatly help us better organize and present the results. All the major changes are highlighted in yellow in the revised manuscript.

As reviewer 1 rightly pointed out, our work is the very first demonstration of yolk-shell architecture for Ge quantum dots in the applications of lithium ion batteries. We have developed a brand-new strategy in synthesizing Ge quantum dot and nitrogen-doped graphene (Ge-QD@NG/NGF) yolk-shell nanoarchitecture. On this basis, we studied the electrochemical performance of the flexible Li-ion batteries with Ge-QD@NG/NGF electrodes. We have also provided evidence that the yolk-shell nanoarchitecture offered ultrahigh performance with utilizing of internal void space for lithiation/delithiation process. This study can be also extended to other high-capacity electrode materials, such as alloys (Sn, and Si) and metal oxides (Co_3O_4 , SnO_2 , and GeO_2), which could ignite interests in a broad readership of Nature Communications, as well as those working with functional flexible graphene structures.

We also thank Reviewer 1 on the suggestion of adding the comparison of the cycle and rate performance of various Ge-based composites prepared by different methods. As shown in Table 1, it has been found that the Ge-QD@NG/NGF electrode offered an ultra-high rate performance as 1001 mAh g^{-1} @ 10C, which is around 2 times larger than that of the carbon-coated Ge nanowires⁴². This electrode also delivered more than 98% specific reversible capacity retention from the second to 1000 cycles of flexible lithium ion battery testing, which is one of the highest known values. Furthermore, the measured specific reversible capacity is 1220 mAh g^{-1} @ 1C, which is again the highest among all the reported works. Therefore, we are confident that our work is of high impact and able to open up unique opportunities for fundamental studies of yolk-shell nanostructures, as well as enable exploration of new functional flexible graphene structures for energy storage research. We have updated this comparison table and discussion into the revised manuscript to provide a direct comparing in terms of performance. Please see **Page 4, line 18-line 25**.

Materials	Structure	Binder/carbon black	Capacity retention	Rate	Capacity (mAh g ⁻¹)	References
Ge	3D macroporous Ge particle structure	10%/10%	96% after 200 cycles	5 C	717	33
Ge	Mesoporous Ge particles structure	15%/15%	99% after 300 cycles	5 C	804	34
C-Ge	Carbon-encapsulated Ge nanowires structure	5%/10%	84% after 50 cycles	10 C	792	35
C-Ge	Carbon-interconnected Ge nanocrystals structure	5%/10%	98% after 1000 cycles	10 C	881	36
CC-Ge	Carbon cubes-encapsulated Ge nanoparticles structure	10%/10%	91% after 500 cycles	30 C	615	37
GTs-Ge	Graphite Tubes-encapsulated Ge nanowires structure	20%/10%	90% after 100 cycles	7 C	232	38
CNT-GNs-Ge	Ge-graphene-carbon nanotube composite structure	10%/10%	81% after 100 cycles	2 C	645	39
GNs-Ge	Graphene-coating Ge nanowires structure	15%/15%	95% after 200 cycles	20 C	363	40
CNF-Ge	Freestanding carbon nanofibers-encapsulated Ge nanoparticles structure	0/0	90% after 100 cycles	15 C	480	41
CNFs-C-Ge	3D flexible carbon-coated Ge nanowires on carbon nanofibers structure	0/0	95% after 100 cycles	10 C	484	42
Ge-QD@NG/NGF	3D flexible nitrogen-doped graphene foam with encapsulated Ge quantum dot@nitrogen-doped graphene yolk-shell structure	0/0	98% after 1000 cycles	10 C 20 C 40 C	1001 905 801	This work

Table 1. A comparison of the cycle and rate performance of various Ge-based composites prepared by different structure designs.

References:

33. Jia, H. P. et al. Reversible storage of lithium in three-dimensional macroporous germanium. *Chem. Mater.* 26, 5683-5688 (2014).
34. Choi, S., Kim, J., Choi, N. S., Kim, M. G. & Park, S. Cost-effective scalable synthesis of mesoporous germanium particles via a redox-transmetalation reaction for high-performance energy

- storage devices. *ACS Nano*. 9, 2203-2212 (2015).
35. Liu, J. et al. Ge/C nanowires as high-capacity and long-life anode materials for Li-ion batteries. *ACS Nano*. 8, 7051-7059 (2014).
36. Ngo, D. T. et al. Carbon-interconnected Ge nanocrystals as an anode with ultra-long-term cyclability for lithium ion batteries. *Adv. Funct. Mater.* 24, 5291-5298 (2014).
37. Li, D., Wang, H. Q., Liu, H. K. & Guo, Z. P. A new strategy for achieving a high performance anode for lithium ion batteries-encapsulating germanium nanoparticles in carbon nanoboxes. *Adv. Energy Mater.* 6, 1501666-1501671 (2016).
38. Sun, Y., Jin, S. X., Yang, G. W., Wang, J. & Wang, C. X. Germanium nanowires-in-graphite tubes via self-catalyzed synergetic confined growth and shell-splitting enhanced Li-storage performance. *ACS Nano*. 9, 3479-3490 (2015).
39. Fang, S., Shen, L. F., Zheng, H. & Zhang, X. G. Ge-graphene-carbon nanotube composite anode for high performance lithium-ion batteries. *J. Mater. Chem. A*. 3, 1498-1503 (2015).
40. Kim, H., Son, Y. K., Park, C., Cho, J. & Choi, H. C. Catalyst-free direct growth of a single to a few layers of graphene on a germanium nanowire for the anode material of a lithium battery. *Angew. Chem. Int. Ed.* 52, 5997-6001 (2013).
41. Li, W. H., Yang, Z. Z., Cheng, J. X., Zhong, X. W., Gu, L. & Yu, Y. Germanium nanoparticles encapsulated in flexible carbon nanofibers as self-supported electrodes for high performance lithium-ion batteries. *Nanoscale*. 6, 4532-4537 (2014).
42. Li, W. H. et al. Carbon-coated germanium nanowires on carbon nanofibers as self-supported electrodes for flexible lithium-ion batteries. *Small*. 11, 2762-2767 (2015).

Comment 2: The methodology followed is reasonable, including synthesis, structural characterisation and electrochemical performance parts. However, it is quite disappointing that it was not thoroughly proof-read before submitting to Nature Communications, as this distracts somewhat from the science. For example, the discussion about the properties (pg. 8-13) is difficult to follow, since it is not presented in a logical order.

Response/corrections: We sincerely appreciate the reviewer's time and important comments. We apologize for the insufficient proof reading in the previous version. We have carefully checked the logistics order and revised our manuscript accordingly. We hope this revised version is acceptable for publication.

Comment 3: Starting with Figure 3(a), continues to 4(a) and 4(b), then goes back to 3(c), 3(e) and 3(f), followed by a jump to Figure 5 (a,b,c,d in an order), back to Figure 3(d), 3(b), then 4(c), 4(d), 4(e) and finishes with 4(f). Either the figures need to change to follow the order of the text, or the text needs to change to reflect the order of the figures. There are also missing parts in the figures, especially in Figure 3, which demonstrates the electrochemical performance of the material and is actually the strong point of this communication, according to the abstract, and hence it should be very well presented. For instance, in Figure 3(c), the y2 axis is presumably columbic efficiency (%) as mentioned in the caption, however this is missing on the figure and there is no connection to the blue line, which is probably corresponding to the coulombic efficiency.

Response/corrections: We appreciate the constructive comments from the reviewer. We have reorganized the figure 3, 4 and 5 carefully according to these suggestions. We found the strong points in the paper have been all well-presented in this manner. The new Figure 3, 4 and 5 with rephrased Figure Captions have been updated as below:

Figure 3. a) Galvanostatic charge-discharge profiles in the 0.01 V-1.5 V window (vs. Li/Li⁺) for the 1st, 2nd, 10th, 100th, and 1000th cycles at 1 C. b) Cycling performance (discharge) and coulombic efficiency of the Ge-QD@NG/NGF/PDMS yolk-shell electrode, Ge/NGF/PDMS and Ge/Cu electrodes at 1 C for 1000 cycles. c) A schematic diagram of the “transparent” half cell for in situ micro-Raman measurement. d) Selected Raman spectra of the half cell during galvanostatic lithiation of the Ge-QD@NG/NGF/PDMS yolk-shell nanoarchitecture at a rate of C/10. A laser power of 2.5 mW and a collection time of 30 s were used for each spectrum, for each acquisition 10 spectra were accumulated. e) and f) Comparison between electrode design in which Ge is coated on an Cu foil and a NGF-based flexible electrode.

Figure 4. a) Typical stress-strain curves of free-standing 3D electrode structure with and without Ge-QD@NG. b) Tensile strength and modulus of free-standing 3D electrode structure with and without Ge-QD@NG. c) Galvanostatic charging/discharging curves of the battery. Red and blue lines represent the as-fabricated flat battery and the bent battery after repeatedly bending, respectively. d) Cyclic performance of the battery under flat and bent states.

Figure 5. a) Rate performance of the Ge-QD@NG/NGF/PDMS yolk-shell, Ge/NGF/PDMS and Ge/Cu electrodes at different current densities. b) Nyquist plots of the Ge-QD@NG/NGF/PDMS yolk-shell electrode after the 1st, 2nd, 10th, 100th and 1000th cycles at a current density of 1 C. c-e) SEM (c) and TEM (d,e) images of Ge-QD@NG/NGF/PDMS yolk-shell electrode in lithium intercalation state after 1000 cycles at a current density of 1 C. f) Schematic drawing of the charge/discharge processes of the Ge-QD@NG/NGF/PDMS yolk-shell electrode.

Comment 4: It should also be mentioned that Figure S6, which is discussed in pg. 12 does not exist!

Response/corrections: We thank the reviewer for this suggestion and apologize for the inadvertent omission in the previous version. The discussion section in Page 12 has been updated. The Figure mentioned in the content is Figure S7 (previous Figure S6), the discussion of this Figure has been added into the revised manuscript.

Page 12, line 10- line 18. “In order to understand the excellent rate capability, the Nyquist plots of Ge/Cu, Ge/NGF/PDMS and Ge-QD@NG/NGF/PDMS yolk-shell electrodes (Figure

S7). Apparently, the Ge-QD@NG/NGF/PDMS yolk-shell electrode displays a much lower resistance than the Ge/Cu and Ge/NGF/PDMS electrode. Significantly, the 3D interconnected porous NGF structural design is proved to be useful in minimizing electron and ion transport resistance. Moreover, the N-doped graphene outershell acts as a channel for lithium ion to Ge during lithium alloying and leaching, which is indicated by the high diffusion coefficient of N-doped graphene compared with germanium.”

Figure S7. Nyquist plots of the Ge-QD@NG/NGF/PDMS yolk-shell electrode (black), Ge/NGF/PDMS (green) and Ge/Cu (red) electrodes.

Comment 5: Moreover, the acronyms are not defined clearly and this can be confusing to a reader who is not familiar with the terminology. For example, the acronym for the polymer used (PDMS, presumably polydimethylsiloxane) is not given at all in the text. Also, it would help if the acronyms, which are used in the text (pg. 5-6) at each step of the synthesis procedure, were indicated within the relevant schematic in Figure 1.

Response/corrections: We thank reviewer 1 for the important suggestions on the definition of acronyms for our manuscript. We have carefully checked the entire manuscript and gave clear definition for all the terminologies. All the corrected acronyms have been highlighted in the revised manuscript.

On description of Figure 1, we have rephrased in **Page 6, line 5 - line 11** as: “As a final step, in order to test the flexibility and electrochemical property of the nanoarchitecture, the thin layer of poly (dimethyl siloxane) (PDMS) was uniform coated on the surface of the Ge-QD@NG/NGF yolk-shell nanoarchitecture (Ge-QD@NG/NGF/PDMS). It is noteworthy that PDMS gel was prepared by intensively stirring base/curing agents (Sylgard 184; Dow Corning), then by degassing in vacuum for 1 h and thermally curing for 6 h under 80 °C.” We have also indicated the relevant acronyms for each step in the schematic in Figure 1. We believe that readers will have a better understanding now.

Figure 1. A schematic illustration of the preparation of Ge-QD@NG/NGF/PDMS yolk-shell nanoarchitecture.

Comment 6: The structural characterisation of the Ge-QD@NG/NGF nano-architecture, is well supported by XRD, TEM, and Raman, in Figure 2. However, it is not clear how the samples for comparison (Ge nanoparticle and Ge/NGF) were made. This could possibly be added in Figure 1 and introduce the acronyms used in the rest of the text.

Response/corrections: We thank the Reviewer for this valuable suggestion. In order to make a fair comparison with meaningful control experiments, the preparations of Ge nanoparticle and Ge/NGF nanoarchitecture have been carefully conducted. We have updated these important information in the Experimental Section in the revised manuscript.

As shown in **Page 16, line 7- Page 17, line 3:**

“Preparation of Ge nanoparticle:

GeCl₄ (99.99%) was dissolved in ethanol to form a solution (0.1 mol L⁻¹) and soaked in above solution before being transferred to a 100 mL Teflon-lined autoclave and hydrothermally treated at 100 °C for 10 h. Then as-synthesized sample was recovered using centrifugation and dried under vacuum chamber overnight further characterization.

Preparation of Ge/NGF nanoarchitecture:

Firstly, the porous N-doped graphene foam structure was synthesized by a CVD technique using the porous Ni foam as the template with pyridine as the nitrogen and carbon sources for N-doped graphene growth. The porous Ni foam was cut into pieces of 7 cm × 4 cm and placed in a quartz tube furnace and annealed at 900 °C in flowing Ar (90%)/H₂ (10%) gas mixture for 20 min to reduce the surface oxide layer. After that, pyridine was decomposed under the mixed atmosphere of Ar (90%)/H₂ (10%) gas for N-doped graphene growth for 5 min under 900 °C. Then, GeCl₄ (99.99%) was dissolved in ethanol to form a solution (0.1 mol L⁻¹). The obtained N-doped graphene with porous Ni foam was soaked in above solution before being transferred to a 100 mL Teflon-lined autoclave and hydrothermally treated at 100 °C for 10 h. The GeO₂/NGF were kept in avacuum oven for 10 min under room temperature, and heated at 650 °C with the protection of Ar (90%) /H₂ (10%) atmosphere for 6 h. After

that, the resulting powder was etched with 1M HCl solution to remove Ni foam. The final samples were kept in a vacuum chamber to avoid oxidation for further characterization.”

Comment 7: The characterisation for Ge nanoparticle (Ge) sample is shown in Figures S1 and S2 and is convincing, although a comment about the estimated size of Ge nanoparticles, as shown in Figure S2, would be necessary. The characterisation for Ge/NGF (Figure S3, which is not discussed at all at the text) is quite poor. TEM would help to indicate the difference with the Ge-QD@NG/NGF nano-architecture. Both for the Ge and Ge/NGF samples, the axis of XRD (Figures S1 and S3) should extent to 90° 2theta degrees for consistency with the XRD pattern for the Ge-QD@NG/NGF (Figure 2h).

Response/corrections: Following Reviewer’s comments, we have reorganized Figure S2 and Figure S3. TEM image in Figure S2 has been used to estimate the size distribution of Ge nanoparticles. It was estimated that Ge nanoparticles were uniformly distributed with an average diameter as ~ 20 nm. As suggested by Reviewer, the axes of XRD (Figures S2 and S3) have been extended to 90° 2-theta degrees for consistency with the XRD pattern for the Ge-QD@NG/NGF (Figure 2h) in the revised manuscript.

The updated Figure S2 and S3 with Figure Captions are as following:

Figure S2. a) XRD pattern and b) TEM image of the Ge.

Figure S3. a) XRD pattern and b) Raman spectra of Ge/NGF nanoarchitecture.

The discussion parts about these two figures have also been updated in **Page 8, line 8- line 16**. “For comparison, the pure Ge nanoparticle (Ge) and 3D interconnected porous

nitrogen-doped graphene foam with encapsulated pure Ge nanoparticles (Ge/NGF) architecture were also synthesized by using the same experimental condition without Ni coating layer (Figure S2a and S3, Supporting Information). It is clear from the TEM image that the Ge nanoparticle without Ni coating layer has been further growth under the thermal reduction process with an average diameter of ~20 nm (see Supporting Information, Figure S2b and S4). It is noted that the presence of the Ni coating layer can restrict the further growth of Ge nanoparticles under the thermal reduction process.”

Comment 8: The Ge content in Ge-QD@NG/NGF nano-architecture was estimated by thermogravimetric analysis (TGA, Figure S4), by using information for the Ge and Ge/NGF samples and the analysis is shown in Table S1. It should be mentioned that it is not clear how the equation used is derived and whether the terms W_{NG} , W_{Ge} , $W_{Ge-QD@NG/NGF}$, X_{Ge} correspond to the tabs in the table.

Response/corrections: We apologized for the confusing terminology in our original manuscript. The terms (i.e., W_{NG} , W_{Ge} , $W_{Ge-QD@NG/NGF}$) are actually used to represent the residual weight percentage of NG, Ge and Ge-QD@NG/NGF at different temperature, respectively. In the computation, X_{Ge} is used to represent the mass ratio of Ge in the table S1. We have updated this information to the revised manuscript. The thermogravimetric analysis equation is derived by Ji et al in 2013¹. We have added this reference in Table S1.

	NGF (wt%)	Ge (wt%)	Ge-NG/NGF(wt%)	$X_{Ge}:X_{NG}$ (mass ratio)
600 °C	0.06	105.12	77.22	73.44 : 26.56
625 °C	0.04	105.74	78.07	73.82 : 26.18
650 °C	0.03	106.39	78.58	73.85 : 26.15
675 °C	0.03	107.05	79.07	73.86 : 26.14
700 °C	0.03	107.13	79.11	73.84 : 26.16

Table S1. Mass ratio of Ge and nitrogen-doped graphene at different temperature¹.

Reference:

1. Ji, J. Y. et al. Graphene-encapsulated Si on ultrathin-graphite foam as anode for high capacity lithium-ion batteries. *Adv. Mater.* 25, 4673-4677 (2013).

Comment 9: Also, the PDMS polymer would be expected to burn off at some temperature for the Ge-QD@NG/NGF sample; this is not mentioned at all (if not, a reference should be given). Finally, the discussion about the TGA results (pg.8) refers to 700 °C, however it is not clear as to where this data point comes from, since in the analysis in Table S1 the maximum temperature is 675°C.

Response/corrections: This material is the Ge-QD@NG/NGF nano-architecture, rather than Ge-QD@NG/NGF/PDMS nano-architecture in the thermogravimetric analysis (TGA, Figure S1, Supporting Information). Therefore, there is no PDMS in the thermogravimetric analysis. Furthermore, the TGA results refers to 700 °C (Table S1, Supporting Information) have been added in the revised manuscript.

Figure S1. TGA curves of Ge, NGF, and Ge-QD@NG/NGF in air gas at a heating rate of 5 °C min⁻¹.

Comment 10: The electrochemical performance is shown well, although some rearrangement of the figures and/or text would be necessary, as aforementioned. However, there is no evidence that the lithiated phase is Li_{4.4}Ge as shown in the schematic in Figure 4(f) and this is not discussed in the text. Figure 4(e) is an important figure, aiming to demonstrate the Li intercalation in the structure and the retention of the yolk-shell architecture after 1000 cycles, by TEM. It would be good to do in-situ TEM to follow the insertion of Li into Ge-QD@NG/NGF, if possible.

Response/corrections: We thank the Reviewer for this valuable suggestion. Please kindly note that we have reorganized the figures as shown in the response to Comment 3. The new Figure 5f (previous Figure 4f) have been carefully discussed in the text.

Please see the updated discussion of Figure 5f in **Page 13, line 14-Page 14, line 6**. “As shown in Figure 5f, the excellent cycle stability and ultra-high rate capability of Ge-QD@NG/NGF yolk-shell nanoarchitecture may be attributed to the following mechanism: i) The electrode materials, Ge, has been homogeneously dispersed with a size down to several nanometers. In such a manner, the compact contact between electrode materials and current collector can be achieved, which is favorable for electrode materials activation processes. It is also believed that quantum-size confinement effect for lithium storage can occur in surface region of electrode materials and therefore improve the specific capacity^{50,51}. ii) The design of yolk-shell structure not only effectively alleviate the huge volume expansion/ contraction during charge-discharge process, but also make the SEI to form mainly on the N-doped graphene outer shell, which has seal any possible defects and prevent electrolyte infiltration through the conformal outer shell²⁴⁻²⁸. iii) The entire 3D interconnected porous electrode system was formed with a well-defined porosity, maximized surface area, and enlarged lattice spacing between graphene layers, coupled with the N-doping-induced defects. Such structure is beneficial for the synergistic effects, which facilitate the fast lithium ion and electron diffusion, improve the storage of lithium ions, and accommodate the effect of huge volume change during the lithiation/delithiation process⁵²⁻⁵⁴.”

We strongly agree with the Reviewer that the use of in situ TEM to follow the insertion of Li ions will better present our novel strategy. Due to the limited resource, we could not include such results in this manuscript in short term. However, we will definitely perform similar experiments in our future work and include in our following publications.

Comment 11: Figure 4(b) shows the results of in-situ Raman during lithiation of Ge-QD@NG/NGF, however is not discussed much in the text and no references are given to support the argument that Li is actually incorporated in the structure. Finally, not much information is given for the PDMS coating, as this is described in the synthesis part (page 6) but not at the experimental section (pg.16). Its role (possibly linked with the observed flexibility?), is not discussed in the text.

Response/corrections: In our current manuscript, we have utilized ex situ TEM combined with in situ Raman spectroscopy to demonstrate the Li intercalation process, which we found very accurate and meaningful at current stage. An important reference⁴⁹ has been added to the manuscript. We have updated a detailed description in **Page 9, line 19-Page 10, line 9**. PDMS coating was used to fully demonstrate the flexibility of our electrodes. As for the information on preparation of PDMS coating, we have described with all the details in the synthesis part (**Page 6 line 5- line 11**) of the revised manuscript.

Reference:

49. Zeng, Z. D. et al. *In situ* measurement of lithiation-induced stress in silicon nanoparticles using micro-Raman spectroscopy. *Nano Energy*. 22, 105-110 (2016).

Comment 12: At this point, I am unsure whether to support this article for publication in Nature Communications. The demonstrated promising performance needs to be compared and discussed in the general context of Ge nano-architectures and other metal containing yolk-shell nano-architectures existing in the literature for Li-ion batteries. It also needs serious re-writing in a reader-friendly manner and some scientific evidence for the claims presented.

Response/corrections: We thank the reviewers again for the constructive suggestions and valuable time to carefully review our manuscript, which significantly improved the quality of this work. We have revised the manuscript to address all these comments with detailed responses. We believe that all the reviewer concerns have been addressed and hope that the manuscript is acceptable for publication.

Response to Reviewer #2:

This manuscript introduced superior rate capability and cycling stability of flexible lithium ion battery anode prepared from 3D interconnected porous nitrogen-doped graphene foam with encapsulated Ge quantum dot@nitrogen-doped graphene yolk-shell nanoarchitecture. By the rational design of novel electrode nanoarchitecture, structure agglomeration and volume change are greatly alleviated, and faster electronic and ionic paths are provided. The idea was cleverly conceived and experiments were carefully carried out. This work would make big impacts to the field of high capacity anode materials with large volume change for flexible lithium ion batteries. The paper is ready to be published after addressing some minor concerns.

Response/corrections: We thank the Reviewer for this encouraging comment and are glad that the reviewer finds our work can be acceptable for publication with minor revisions. All concerns have been taken into consideration and addressed in the revised manuscript. We hope that the Reviewer finds the revised manuscript suitable for publication now.

Comment 1: The authors need to clarify what weight basis of the specific capacity is calculated on. Is it based on the total weight of the nanoarchitecture or based on the Ge quantum dot only? Without this information, it is hard to compare the performance with other reported one.

Response/corrections: In this work, the specific capacity was calculated based on the total mass of the composite including the graphene and germanium. We thank the Reviewer for highlighting this important point. This information has been updated in the revised manuscript on page 18.

Page 17, line 24-Page 18, line 1. “It's remarkable that all the electrochemical data was calculated based on the total mass of the composite including graphene and germanium in this work.”

Comment 2: Can the author give the specific surface area of the as-prepared Ge-QD@NG/NGF yolk-shell nanoarchitecture?

Response/corrections: The specific surface area of the as-prepared Ge-QD@NG/NGF yolk-shell nanoarchitecture have been evaluated to be $\sim 392.8 \text{ m}^2\text{g}^{-1}$, which has been added in the revised manuscript (see Supporting Information, Figure S5).

Please see **Page 8, line 16-line 20.** “In addition, a total specific surface area of $392.8 \text{ m}^2 \text{ g}^{-1}$ was obtained by the Brunauer-Emmett-Teller (BET) method (see Supporting Information, Figure S5). Such high surface area provides more surface-active sites and makes the diffusion of the liquid electrolyte into the electrodes more easily, leading to an enhancement of the electrochemical performance.”

Figure S5. Nitrogen adsorption/desorption isotherms of the Ge-QD@NG/NGF yolk-shell nanocomposite.

Comment 3: The authors need to provide the information on the total weight of the materials when fabricated as electrodes. Too little material will cause much error when calculating the specific capacity.

Response/corrections: We thank the reviewer for this comment and agree that indeed the active mass loading of the electrode should be added. We have provided this information in the revised manuscript.

Please see **page 17, line 17- line 23**: “In particular, the mass loading of electrode materials was calculated at ca. 1.8 mg cm^{-2} and assembled into two-electrode CR2025-type coin cells in an Ar-filled glove box for next battery measurements, porous polypropylene film was used as the separator, lithium foil as the counter electrode. 1 M solution of LiPF_6 in a volumetric ratio of 1:1:1 mixture of ethylene carbonate (EC) and dimethyl carbonate (DMC) and diethyl carbonate (DEC) was used as an electrolyte.”

Comment 4: The authors should pay particular attention to English grammar and sentence structure to make the description be clear to the readers.

Response/corrections: We thank the Reviewer for this valuable comment. We have revised the manuscript carefully with improved English grammar and sentence structure.

Reviewers' comments:

Reviewer #1 (Remarks to the Author):

The revised manuscript addresses quite well the main concerns indicated in the first manuscript.

Comment 1: Table 1 is comprehensive and enables direct comparison with other Ge-based composites. The performance at 1C should also be included, since this is the one mainly described in the text and abstract (1220mAh/g). The capacity retention of 98% after 1000 cycles is only demonstrated for rate 1C (Fig. 3b) and for 200cycles for higher rates (10C, 20C, 40C in Fig S6); this should be definitely indicated in the table.

Comments 2, 3 and 4: The ordering of the figures now follows the text and hence it is much more reader-friendly than in the first version of the manuscript.

Comment 5: The acronyms, most importantly for Fig. 1 (synthesis steps), are now clearly given and this enables readers to follow the text easily.

Please provide details:

What is the 'appropriate void space' defined and how can this be 'tuned'? What was the thickness of the Ni sacrificial layer in Ge-QD@Ni/NG-NF?

Comments 6: The experimental details for the preparation of the two materials used for control experiments (Ge nanoparticle and Ge/NGF nanoarchitecture) have been added.

Comment 7: Could the authors comment on whether the enhanced performance of Ge-QD@NG/NG-NF/PDMS compared to Ge/NGF nanoarchitecture, as shown in Fig. 3a and 3b, is mainly the result of the void space (thanks to the Ni sacrificial layer in Ge-QD@NG/NG-NF/PDMS) or the size of Ge?

Could the authors provide a TEM image for Ge/NGF nanoarchitecture? This would be necessary to demonstrate the differences in morphology with Ge-QD@NG/NG-NF/PDMS.

Comments 8 and 9: OK

Comments 10: The added text discusses well the demonstrated promising behavior of Ge-QD@NG/NG-NF/PDMS.

Comment 11: Is the thickness and porosity of the PDMS layer controllable? And if so, is it the same for both Ge-QD@NG/NG-NF/PDMS and Ge/NGF/PDMS? If not, what would be the effect on the properties?

Regarding the bending behavior (Figure 4): For Fig. 4c: At which cycle number in Fig. 4d does it correspond and at which current density? For Fig. 4d: Please add details in the figure caption (e.g. the grey points and the red line). Also, provide experimental details on how the bending test was performed.

Comment 12: I would be happy to support this manuscript for publication in Nature Communications, after addressing the highlighted concerns (comments 1,5,7,11).

Reviewer #2 (Remarks to the Author):

As the authors have carefully and fully addressed the comments I raised, this work can be accepted now.

Reviewers' comments:

Reviewer #1 (Remarks to the Author):

The revised manuscript addresses quite well the main concerns indicated in the first manuscript.

Response/corrections: We sincerely appreciate the reviewer's time and important comments to review our original and revised manuscripts. We have carefully studied the comments and revised our manuscript accordingly. All the major changes are highlighted in yellow in the revised manuscript. We hope this revised version is acceptable for publication.

Comment 1: Table 1 is comprehensive and enables direct comparison with other Ge-based composites. The performance at 1C should also be included, since this is the one mainly described in the text and abstract (1220mAh/g). The capacity retention of 98% after 1000 cycles is only demonstrated for rate 1C (Fig. 3b) and for 200cycles for higher rates (10C, 20C, 40C in Fig S6); this should be definitely indicated in the table.

Response/corrections: We thank the Reviewer for this valuable suggestion. We have added the performance at 1C and updated this comparison table to provide a direct comparing in terms of performance in the revised manuscript.

Materials	Structure	Binder/carbon black	Capacity retention	Rate	Capacity (mAh g ⁻¹)	References
Ge	3D macroporous Ge particle structure	10%/10%	96% after 200 cycles at 1 C	5 C	717	33
Ge	Mesoporous Ge particles structure	15%/15%	99% after 300 cycles at 0.5 C	5 C	804	34
C-Ge	Carbon-encapsulated Ge nanowires structure	5%/10%	84% after 50 cycles at 0.2 C	10 C	792	35
C-Ge	Carbon-interconnected Ge nanocrystals structure	5%/10%	98% after 1000 cycles at 0.5 C	10 C	881	36
CC-Ge	Carbon cubes-encapsulated Ge nanoparticles structure	10%/10%	91% after 500 cycles at 0.5 C	30 C	615	37
GTs-Ge	Graphite Tubes-encapsulated Ge nanowires structure	20%/10%	89% after 100 cycles at 0.2 C	7 C	232	38
CNT-GNs-Ge	Ge-graphene-carbon nanotube composite structure	10%/10%	81% after 100 cycles at 0.1 C	2 C	645	39
GNs-Ge	Graphene-coating Ge nanowires structure	15%/15%	95% after 200 cycles at 0.5 C	20 C	363	40

CNF-Ge	Freestanding carbon nanofibers-encapsulated Ge nanoparticles structure	0/0	90% after 100 cycles at 0.15C	15 C	480	41
CNFs-C-Ge	3D flexible carbon-coated Ge nanowires on carbon nanofibers structure	0/0	95% after 100 cycles at 0.1 C	10 C	484	42
Ge-QD@NG/NGF	3D flexible nitrogen-doped graphene foam with encapsulated Ge quantum dot@nitrogen-doped graphene yolk-shell structure	0/0	98% after 1000 cycles at 1 C	1 C	1220	This work
			97% after 200 cycles at 10 C	10 C	1001	
			97% after 200 cycles at 20 C	20 C	905	
			98% after 200 cycles at 40 C	40 C	801	

Table 1. A comparison of the cycle and rate performance of various Ge-based composites prepared by different structure designs.

Comment 5: The acronyms, most importantly for Fig. 1 (synthesis steps), are now clearly given and this enables readers to follow the text easily. Please provide details: What is the ‘appropriate void space’ defined and how can this be ‘tuned’? What was the thickness of the Ni sacrificial layer in Ge-QD@Ni/NG-NF?

Response/corrections: We thank the Reviewer for this valuable suggestion to improve the quality of this paper. The key design of the yolk-shell nanoarchitecture in improving electrochemical performance lies in the appropriate void space, which would be expand/contract freely upon lithium alloying and leaching without damaging the outer shell. In this work, this Ni sacrificial layer is very critical because it determines the internal void space (that is, Ni sacrificial layer) between the NG shell and Ge-QD core. If the Ni sacrificial layer is very thin, it may crack or disintegrate upon lithiation/delithiation process result in the rapid capacity fade of the electrode. Therefore, the Ni thin layer serves as not only the catalyst for N-doped graphene growth, but also the sacrificial coating layer for providing internal void space. In this work, the thickness of the Ni sacrificial layer is ~2.2 nm internal void space in Ge-QD@Ni/NG-NF. And the thickness of the Ni coating layer may be easily tuned by changing the electroplating deposition parameters.

Please see **Page 5, line 22- Page 6, line 3:** “In this process, we uniform coated GeO₂ nanoparticles with Ni coating layer (GeO₂@Ni) core-shell structure. And thickness of the Ni coating layer may be easily tuned for the appropriate void space by changing the electroplating deposition parameters. It is noted that the key design of the yolk-shell nanoarchitecture in improving electrochemical performance lies in the appropriate void space,

which would be expand/contract freely upon lithium alloying and leaching without damaging the outer shell.”

Please see **Page 13, line 11- line 19**: “TEM image of a lithiated yolk-shell nanoarchitecture shows a distinct N-doped graphene coating (Figure 5e), indicating the Ge-QD core not only may expand/contract freely during lithium alloying and leaching without damaging the N-doped graphene outer shell. This result clearly demonstrates that the incorporation of the yolk-shell structure and 3D interconnected porous graphene-based electrode architectures can accommodate the significant volume expansion/contraction in Li-Ge alloying and de-alloying reactions during the charge-discharge processes, which is the key factor for a high reversible capacity, excellent cycle stability and ultra-high rate capability electrode.”

Comment 7: Could the authors comment on whether the enhanced performance of Ge-QD@NG/NG-NF/PDMS compared to Ge/NGF nanoarchitecture, as shown in Fig. 3a and 3b, is mainly the result of the void space (thanks to the Ni sacrificial layer in Ge-QD@NG/NG-NF/PDMS) or the size of Ge?

Response/corrections: We thank the Reviewer for this valuable suggestion. From our results and observation, we conclude that the enhanced performance is due to two reasons: 1) The electrode materials, Ge, has been homogeneously dispersed with a size down to several nanometers, which prompt a compact contact between electrode materials and current collector. It is also believed that quantum-size confinement effect for lithium storage can occur in surface region of electrode materials and therefore improve the specific capacity; 2) The design of yolk-shell structure not only effectively alleviate the huge volume expansion/contraction during charge-discharge process, but also make the SEI to form mainly on the N-doped graphene outer shell, which has seal any possible defects and prevent electrolyte infiltration through the conformal outer shell. The details to comment on the enhanced performance of Ge-QD@NG/NG-NF/PDMS compared to Ge/NGF nanoarchitecture have been added revised manuscript.

Please see **Page 11, line 2- line 6**: “However, on one hand, the large size of Ge nanoparticles in the Ge/NGF/PDMS electrode has led to poor strain relaxation. On the other hand, the Ge/NGF/PDMS electrode does not provide internal void space to alleviate the huge volume changes of Ge during lithium alloying and leaching, which resulted in a gradual capacity decline in the first 200 circles.”

Please see **Page 13, line 19- Page 14, line 5**: “As shown in Figure 5f, the excellent cycle stability and ultra-high rate capability of Ge-QD@NG/NGF yolk-shell nanoarchitecture may be attributed to the following mechanism: i) The electrode materials, Ge, has been homogeneously dispersed with a size down to several nanometers. In such a manner, the compact contact between electrode materials and current collector can be achieved, which is favorable for electrode materials activation processes. It is also believed that quantum-size confinement effect for lithium storage can occur in surface region of electrode materials and

therefore improve the specific capacity^{50,51}. ii) The design of yolk-shell structure not only effectively alleviate the huge volume expansion/ contraction during charge-discharge process, but also make the SEI to form mainly on the N-doped graphene outer shell, which has seal any possible defects and prevent electrolyte infiltration through the conformal outer shell²⁴⁻²⁸.”

Could the authors provide a TEM image for Ge/NGF nanoarchitecture? This would be necessary to demonstrate the differences in morphology with Ge-QD@NG/NG-NF/PDMS.

Response/corrections: We thank the Reviewer for this valuable suggestion. The TEM image of the Ge/NGF nanoarchitecture has been added in the revised manuscript (see Supporting Information, Figure S4).

As shown in **Page 8, line 14- line 17**: “It is clear from the TEM image that the Ge nanoparticle without Ni coating layer has been further growth under the thermal reduction process with an average diameter of ~20 nm (see Supplementary Fig. 2b and 4).”

Figure S4. a-d) EDS elemental maps of Ge, C, and N, respectively. e) TEM image of the Ge/NGF nanoarchitecture. f) The electronic diffraction pattern corresponding to the Ge.

Comment 11: Is the thickness and porosity of the PDMS layer controllable? And if so, is it the same for both Ge-QD@NG/NG-NF/PDMS and Ge/NGF/PDMS? If not, what would be the effect on the properties?

Response/corrections: In order to fully control the PDMS layer thickness, we have weighed carefully the same quality of poly (dimethyl siloxane) (PDMS) by spin coating to uniform thin layers on the surface of the Ge-QD@NG/NGF and Ge/NGF samples. Both the thickness and porosity of the PDMS layer are controllable by modifying the quality of PDMS and the speed/time of spin coating machine. Therefore, PDMS layers are the same for both Ge-QD@NG/NG-NF/PDMS and Ge/NGF/PDMS samples.

As shown in **Page 9, line 1- line 5**: “In order to test the flexibility and electrochemical property of the nanoarchitecture, we weighed the same quality of PDMS by spin coating to uniform coat on the surface of the Ge-QD@NG/NGF yolk-shell nanoarchitecture (Ge-QD@NG/NGF/PDMS) and Ge/NGF nanoarchitecture (Ge/NGF/PDMS).”

Regarding the bending behavior (Figure 4): For Fig. 4c: At which cycle number in Fig. 4d does it correspond and at which current density? For Fig. 4d: Please add details in the figure caption (e.g. the grey points and the red line). Also, provide experimental details on how the bending test was performed.

Response/corrections: We thank the Reviewer for this valuable suggestion. Figure 4c shows the galvanostatic charging/discharging curves of the battery at charge-discharge rate of 1 C. Red and blue lines represent the as-fabricated batteries under flat state after 20 cycles and the batteries under bending state after 20 cycles (i.e., under repeatedly bending with an angle of 90°) (Figure 4c). We have provided experimental details on how the bending test was performed and added the details in the figure caption (e.g. the grey points and the red line) in the revised manuscript.

As shown in **Page 11, line 20- Page 12, line 2**: “Compared with that of the original flat state, only a negligible over-potential and a less than 2% specific reversible capacity reduction can be observed under bend state (bending angle of 90°) (Figure 4c). Moreover, the Ge-QD@NG/NGF yolk-shell electrode exhibited outstanding cycle stability under flat and bent states. In particular, the Ge-QD@NG/NGF/PDMS yolk-shell electrode exhibited a reversible specific capacity retention of ~96.8% from the second to first 20 cycles under a flat state, and ~95.2 % after another 20 cycles under a bent state at charge-discharge rate of 1 C (Figure 4d).”

Figure 4. a) Typical stress-strain curves of free-standing 3D electrode structure with and without Ge-QD@NG. b) Tensile strength and modulus of free-standing 3D electrode structure with and without

Ge-QD@NG. c) Galvanostatic charging/discharging curves of the battery at charge-discharge rate of 1 C. Red and blue lines represent the as-fabricated flat battery after 20 cycles and the bent battery after 20 cycles under repeatedly bending (bending angle of 90°), respectively. d) Cyclic performance of the battery under flat and bent states. Red and grey lines represent the charging and discharging performance at charge-discharge rate of 1 C.

Comment 12: I would be happy to support this manuscript for publication in Nature Communications, after addressing the highlighted concerns (comments 1,5,7,11).

Response/corrections: We sincerely appreciate the Reviewer's time and encouraging and insightful comments. We have carefully checked the highlighted comments and revised our manuscript accordingly. We hope this second revised version is acceptable for publication.

Reviewer #2 (Remarks to the Author):

As the authors have carefully and fully addressed the comments I raised, this work can be accepted now.

Response/corrections: We appreciate the Reviewer's efforts for carefully reviewing our original and revised manuscript and are glad that the reviewer finds the revised manuscript acceptable for publication.

REVIEWERS' COMMENTS:

Reviewer #1 (Remarks to the Author):

As the points raised have now been satisfactorily addressed, I would be happy to support the revised manuscript for publication in Nature Communications.